# *Cis*-Regulation by *NACs*: A Promising Frontier in Wheat Crop Improvement

**DOI:** 10.3390/ijms232315431

**Published:** 2022-12-06

**Authors:** Adnan Iqbal, Joanna Bocian, Amir Hameed, Waclaw Orczyk, Anna Nadolska-Orczyk

**Affiliations:** Plant Breeding and Acclimatization Institute—National Research Institute, Radzikow, 05-870 Blonie, Poland

**Keywords:** wheat, cereals, *TaNAC*, *cis*-regulatory elements, *cis*-elements, yield-related traits, biotic and abiotic stresses

## Abstract

Crop traits are controlled by multiple genes; however, the complex spatio-temporal transcriptional behavior of genes cannot be fully understood without comprehending the role of transcription factors (TFs) and the underlying mechanisms of the binding interactions of their *cis*-regulatory elements. NAC belongs to one of the largest families of plant-specific TFs and has been associated with the regulation of many traits. This review provides insight into the *cis*-regulation of genes by wheat *NAC*s (*TaNAC*s) for the improvement in yield-related traits, including phytohormonal homeostasis, leaf senescence, seed traits improvement, root modulation, and biotic and abiotic stresses in wheat and other cereals. We also discussed the current potential, knowledge gaps, and prospects of *TaNAC*s.

## 1. Introduction

Wheat (*Triticum aestivum*) is one of the main staple food crops cultivated around the world. Although it is a very important cereal crop, it is underexplored compared to rice, barley, and maize. The main reason is its large and complex hexaploid genome, AABBDD (2n = 6x = 42), which has three homoeologous sets of genes that make research more demanding [1,2,3]. The genome of rice is diploid, 33 times smaller than that of wheat; therefore, this species is very well explored and treated as a model among cereals. Barley and maize are diploid species. However, the first one is more closely related to wheat, thus reports on the cis-regulation of genes in these two species, rice, and barley, by NAC TFs are also discussed and compared with wheat.

Plant growth, development, and adaptation to the environment depend on transcriptional regulation mediated by transcription factors (TFs), which play a central role in gene networking. TFs serve two main functions. One, to find suitable DNA-binding sites in DNA, and the other, to recruit other regulatory proteins necessary for the transcriptional cascade [4]. Some of the gene transcription networks are highly conserved, while other gene networks evolved over time. Plant TFs evolve by gene duplication, leading to the functional redundancy of diversification of TFs and, consequently, to the conservation or evolution of the gene transcriptional network [5,6]. Based on the molecular mechanism, the evolution of TFs is categorized into four types: either changes within or outside of the DNA binding domain (DBD) of TFs, or changes upstream or downstream of the gene regulatory network (GRN) of TFs. Changes in cis-regulatory elements are the main causes of GRN changes that are the main component of gene expression level [5,7].

TFs constitute about 7% of the total transcriptome [8]. According to plant transcription factor binding databases, more than 300,000 TFs have been reported in plants. They are known to regulate many transcriptional reactions by interacting with *cis*-elements of the gene of interest. Some of the prominent families of plant TFs include NAC, WRKY, MYB, bZIP, and bHLH, which mainly regulate biotic and abiotic stress in plants [9,10,11,12]. Generally, a TF consists of four parts: (i) DNA binding composed of conserved residues, (ii) a non-conserved transcriptional regulatory part, (iii) a dimerization/oligomerization part, and (iv) a nuclear localization signal-containing part (NLS). Most TFs involved in stress, including the regulation of biotic and abiotic stress, have been well documented [13].

NAC is one of the largest families of plant-specific TFs. NAC is the abbreviation of three different genes (*NAM-Non Apical Meristem*; *ATAF-Arabidopsis Transcription Activation Factor*; *CUC-Cup-Shaped Cotyledon*). According to data for *NAC* genes in wheat, available in the Plant Transcription Factor Database version 4, 263 of them have been reported, making it one of the largest individual representatives with plant-specific NAC proteins after soybean [13]. NAC proteins act as TFs and regulate many biological and physiological functions in plants by interacting with cis-elements in the promoter regions of many genes [12]. NAC TFs have been reported to control morphogenesis, senescence, biotic and abiotic stresses, and plant hormone homeostasis [8,14,15]. The role of NAC TFs in the regulation of plant functions in non-stress conditions is still under-discovered; only functions of several TaNAC in wheat have been reported.

More and more research indicates that *NAC* genes are important regulators of yield-related traits. It refers to the direct regulation of yield parameters such as seed-associated traits. However, they might also be involved indirectly in yield improvement by the regulation of phytohormonal homeostasis, especially in generative organs, root development, leaf senescence and/or biotic and abiotic stresses. All these topics are reviewed and discussed.

The knowledge gained through the exploration of the role of TFs and their binding sites on *cis*-elements could be used to improve crop yield and productivity. This review focuses on recent developments in *cis*-regulation of wheat genes by NAC TFs and their role in crop improvement, their current potential, knowledge gaps, and prospects. The review also emphasizes the use of biotechnological tools to cope with the future challenges of food security by enhancing crop yield.

## 2. Structural Attributes of NAC

Generally, NAC proteins are divided into two parts: a well-conserved N-terminal DNA binding domain and a variable C-terminal transcriptional regulatory region [13]. The conserved N-terminal domain is further sub-grouped into five A–E subdomains. Subdomains A, C, and D are conserved in most plant species, while B and E are relatively less conserved. Subdomain A may be associated with protein dimerization, while the variability in subdomains B and E involves the diversity of NAC genes [16,17]. Subdomains C and D contain nuclear localization signals (NLS) in most of the NAC TFs, enabling them to localize in the nucleus and allowing the identification of cis-elements in a gene promoter region. However, the D and E subdomains can physically bind to DNA [18]. These subdomains are characterized for some essential residues for DNA binding; for example, at the positions Lys-123 and Lys-126 in subdomain D, Val-119 to Ser-183 in subdomains D to E, while in subdomain C Arg-85 and Arg-88. However, among all NACs, Arg-88 has been found relatively to be the most conserved amino acid [12,13,16,19]. The C-terminal regulatory part of the NAC protein is variable due to repeated acid residues such as serine, threonine, glutamine, and proline; this variability allows the C-terminal to interact with various targets (Figure 1) [17,20,21]. To date, there are still gaps in understanding the exact functions associated with these NAC subdomains and their expression patterns.

## 3. *Cis*-Regulation

Transcription factors (TFs) are considered very vital in determining cell fate, including cell differentiation, cell development, and responses to environmental stresses by interacting with *cis*-regulatory elements or *cis*-elements (CREs/CEs). CEs are motifs composed of 4–30 bp and distributed in DNA that act as DNA binding sites for TFs, and this interaction is very site specific. Sequential and systematic interaction of TFs with their CEs controls the transcriptional regulatory machinery for a targeted outcome of selected genes [22]. CEs can regulate gene transcription by directly binding TFs to the core promoter or by proximity interaction. In proximity gene regulatory interactions, the site-specific activator or repressor binds to the proximal or distal regulatory sites and interacts with the core promoter. CEs can be present from a few hundred to kilos or even millions of base pairs away at the distal sites of a targeted gene and gene regulation is governed by the spatial interaction of these sites with the core promoter by chromatin coiling and looping (Figure 2) [23,24,25]. CEs are distributed in the proximal, distal, and sometimes intronic regions of the gene, and binding to their site-specific TFs can positively or negatively regulate gene expression [25,26].

The binding of TFs to CEs requires steric stabilization; most CEs bind to nucleosome-free DNA in chromatin, which is known as Accessible Chromatin Regions (ACRs). Genome-wide studies have shown that numerous sequences can be designated as motif-containing sequences or CEs; however, only a small fraction of these CEs (nearly 1%) can bind to their specific TFs and can establish a stable TF-CE interaction [27,28]. This uncovers the fact that only the motif sequence information is not enough to report a stable interaction between TF and its CE sites. Over time, multiple studies have shown that many other factors are involved in TF-CE interactions, such as the identified target sites, the proximal or distal structural features of the sequence to the core DNA binding sites, the ambience of chromatin, the combined action of TF together with other cofactors, and finally the 3-D stretch of DNA in the nucleus [28,29,30,31].

Several in vitro and in vivo techniques have been reported to find plant TFs and their binding sites. The yeast one hybrid assay, the protein binding microarray (‘PBM’), and DNA affinity-purification-sequencing (DAP-seq) are some examples of in vivo techniques, while chromatin immunoprecipitation sequencing (ChIP-seq), (ChIP-exo), (ChIP-nexus) and some methods based on nucleases, including DamID, CUT-RUN, and CUT-TAG are some examples of recent in vitro techniques [32,33,34].

Although a lot of data regarding TFs and their CEs are available, we still lack an understanding of the dynamics of a stabilized TFs–CEs interaction. The lack of knowledge about CEs, including where, when, and which factors activate these regulatory sites distributed in the genome, is another challenge to grasping the cis-regulatory mechanisms [35].

Recent advances in the methods have enabled the identification of the location and characterization of CEs in a single cell. Potential CREs can be located using Assay for Transposase Accessible Chromatin Sequencing (ATAC-seq), and DNase-I hypersensitive sequencing methods for chromatin profiling [36,37,38]. Although recent techniques have enabled locating potential CEs in the genome, unbiased approaches are still required to find/locate active CEs and measure the accurate activity of CEs in plant genomes. Moreover, the problem is to obtain stable cell lines in plants, which are necessary to characterize Ces in a single cell [22].

The *TaSPR* gene from bread wheat encodes NAC protein and it binds to the 5′-CANNTG-3′ CEs distributed in the promoter regions of *SSP* genes encoding seed storage proteins [3]. Similarly, the TaNAC19-A1 NAC protein binds to the 5′-ACGCAG-3′ CEs in the promoter regions of *TaAGPS1-A1* and *TaAGPS1-B1* [39]. Recent studies regarding TaNACs only focus on finding their CEs distributed in the promoter regions of their targeted genes; however, there is a huge gap in knowledge in finding the CEs in proximal and distal locations of the targeted gene. Exploration of steric stabilization of TaNACs to their CEs is another neglected area of research which needs to be focused on further.

## 4. Yield-Related Traits in Wheat and Other Cereals

Yield is one of the most important characteristics of any crop. Therefore, it is important to understand the mechanisms of regulation of yield-related traits by targeting those traits and their driving genes, using breeding and biotechnological approaches. In barley and wheat crops, the inflorescence (classified as a spike) determine the grain weight and grain number, which are the two factors determining the yield. The first trait is calculated by the weighting of 1000 grains and is abbreviated as a thousand-grain weight (TGW). The other is the grain numbers per spike and the spike number per area [40,41,42]. Spikelets are defined as the grain-bearing florets around the spike. In wheat, the spikelets are indeterminate, whereas the spikes are determinate. Unlike wheat, barley has determinated spikelets and undeterminated spikes [41,43]. Compared to wheat and barley, rice has an inflorescence that is classified as a panicle, each spikelet contains one floret, and each floret contains one grain. In rice, the yield is also determined by TGW [40]. The overall yield and grain numbers in wheat, barley, and rice crops depend on many factors, such as inflorescence morphology, tiller numbers, differentiation, vegetative and reproductive phase time, spike and spikelet initiation, elongation, and maturation. Important factors for grain weight are the number of cell divisions in the grain and the sink capacity [42,44,45]. Additionally, crop yield faces several environmental challenges, and the use of functional features of *TaNACs* can help to improve this characteristic.

In earlier research, TaNACs have been documented to be involved in the regulation of important agronomic traits [46]. Uauy et al. [47] found that Gpc-B1, a quantitative trait locus (QTL) of elevated grain protein, is a NAC TF (annotated as NAM-B1), which also accelerates leaf senescence and increases nutrient remobilization. He et al. [46] found that overexpression of nitrate-inducible wheat *TaNAC2-5A* increased root growth, nitrate uptake rate, and grain yield. The allele was also found to bind to the promoter region of genes that encode the nitrate transporter and glutamine synthase, resulting in an improved ability of roots to acquire nitrogen and increased grain yield. Therefore, *NAC*-encoding genes represent another very important resource for more efficient nitrate use and higher-yielding crops [48,49]. More recent studies on the regulation of yield-related traits by *NACs* are summarized below.

## 5. Functional Features of *TaNACs*

Plant-specific NAC TFs have been reported to be associated with many plant functions such as regulation of phytohormonal homeostasis, leaf senescence, abiotic and biotic stresses, the content of seed storage proteins, roots modulation, seed germination, seed setting and vigor, and starch synthesis. Here, we discuss some of the important functional features of TaNAC TFs in detail.

### 5.1. Regulation of Phytohormonal Homeostasis by NAC TFs

Phytohormones are very significant signaling molecules in plant development [50]. In cereals, both groups of main hormones, cytokinins and auxins, are necessary for gametic and somatic embryogenesis and plant development/regeneration [31,51]. In addition to auxins and cytokinins, ABA plays a pivotal role in the development of wheat grains. The higher concentration of cytokinin in the early development of the wheat kernel and the higher concentration of ABA in the later stages resulted in higher yield [52]. Similarly, barley relationships between cytokinins, auxins, and gibberellins are known to regulate spike and spikelet development, which ultimately corresponds to the final yield [53].

Cytokinins generally act by modulating gene transcription in target tissues. As Shanks et al. [54] suggested, the diversity of cytokinin functions is regulated by activities of various TFs altering specific sets of target genes. Members of the large family of NAC TFs have been documented to be involved in the regulation of genes and traits. The expression profiles of 46 *HvNACs* in various barley organs and under two hormone treatments suggested conserved functions of these genes in secondary cell wall biosynthesis, leaf senescence, root development, seed development, and hormone-regulated stress responses. *TaNACs* have also been documented to be involved in the regulation of other important agronomic traits [46,55].

Several studies have shown that *NAC* genes have been induced in many plant species by the exogenous application of phytohormones, including ethylene (ET), jasmonic acid (JA) and salicylic acid (SA). *NAC* genes also regulate the transcription of many stress-related genes by interacting with ABA-dependent and independent signaling pathways [8,56]. Regulation of phytohormonal homeostasis by NAC TFs is an area of great interest, as it plays a basic role in plant development, including yield-related traits in particular environments. In recent studies, its activity in the transcriptional regulation of cytokinin and auxin in rice has been reported. Overexpression of *OsNAC2* in rice plants affected auxin and cytokinin-regulating genes [57]. Molecular analysis revealed that *OsNAC2* can directly bind to the *cis*-elements in the promoter region of the *GH3.6* and *GH3.8* genes, which are IAA inactivation genes. *OsNAC2* also binds to IAA activation genes such as *OsARF25* in rice and modulates auxin hormone levels. In addition to auxin-related genes, *OsNAC2* also regulates cytokinin by directly interacting with *OsCKX4*, a cytokinin oxidase gene [57]. A recent study by Jablonski et al. [58], reported that wheat-selected cytokinin oxidase genes (*TaCKX*) were significantly correlated with *TaNAC2-5A.* Furthermore, the level of expression of this gene was correlated with some yield-related traits. *TaNAC2-5A* was previously described by He et al. [46] as nitrate-inducible, increasing root growth, rate of nitrate uptake, and grain yield.

Phytohormonal homeostasis regulated by NAC proteins could also play an important role in leaf senescence (described below). Numerous studies have reported that most phytohormone-related genes were differentially expressed depending on the stage of this developmental process. Leaf senescence was positively regulated by ET, JA, abscisic acid (ABA), SA, and strigolactones (SLs), while negatively regulated by auxins, cytokinins, and gibberellic acid [59]. The crosstalk between phytohormonal homeostasis and NAC TFs in the wheat crop is underexplored and, to date, not much data is available; therefore, there is a huge knowledge gap that needs to be filled in this area.

### 5.2. Leaf Senescence

Leaf senescence is a complex multifactorial trait that is primarily controlled by genetic and environmental factors. Stay-green is a terminology used for ‘delayed senescence’ that is associated with stress tolerance and, as a consequence, hypothetically longer photosynthetic activity resulting in extended grain filling [60]. In wheat plants, monocarpic senescence is a condition in which most nutrients, along with almost 80% nitrogen (N), are remobilized from the leaves to the grains [60]. Nitrogen remobilization is an important factor in determining the stay-green feature of the faster senescence of plants. Although in plants inefficient remobilization of N from leaves to grains results in delayed senescence and, therefore, has the potential for greater productivity, while efficient mobilization of N causes faster leaf senescence with higher leaf protein content and lower grain yield [61,62]. The remobilization of nutrients accumulated in the leaves to the developing seeds is accompanied by a decrease in CKs in the leaves. Lara et al. [63] reported that the process can be reversed by applying exogenous CK or by an increase in endogenous CK, which delayed senescence and caused nutrient mobilization. As reported by Christiansen et al. [64], barley *HvNAC005* was a strong positive regulator of senescence. In rice, *OsNAC2* promoted leaf senescence by inducing ABA biosynthetic genes and negatively regulating the ABA catabolic gene. *OsNAC2* was also involved in the regulation of chlorophyll degradation genes [65]. These findings indicate that the timing of senescence is a limiting factor for crop yield and certain *NAC*-encoding genes are strong senescence regulators.

The *NAC1* type (*TaNAC-S*) is another type of wheat TF that negatively regulates leaf senescence but interestingly may increase not only grain yield but also grain protein content [66]. *TaNAC-S* has three homologous copies of each of the chromosomes (A, B, and D). *TaNAC-S-A1* was preferentially expressed during flowering days and was found to be primarily associated with grain yield and higher chlorophyll content. Its homologous copy on the B genome, *TaNAC-S-7B2*, was found to be related to grain protein content, but the correlation was not significant. Genetic ontology and gene networking analysis revealed that some other members of the NAC gene family have expression patterns similar to *TaNAC-S* during senescence and nitrogen concentration. They were: *TaNAC29-2A*, *TaNAC-34-2A*, *TaNAC-35-2A,* and *JUB-1* as NAC (*JUB-1* homolog on chromosome 5). However, compared to the *TaNAC-S* expression pattern, some of the genes had opposite transcriptional behaviors, showing a positive relationship with senescence and nitrogen concentration. There were *TaNAC9-2B*, *TaNAC-23-2A*, *TaNAMB2-2B*, and *TaNAMD2-2D* that included some ortholog genes *AtNAP*, *ORS*, *ORE1,* and *NAM-2* [67].

Chlorophyll content has also been reported to have a relationship with delayed senescence. Its altered or prolonged catabolism resulted in the delay of senescence [68]. A wheat transcription factor, *TaNAC2-5A*, has been reported to regulate nitrate concentration and remobilization leading to an increase in yield and seed vigor by binding directly to the *cis*-elements in the promoter region of the *TaNRT2.5* gene, located on the 3B chromosome [46,69]. Down-regulation of *TaCKX2* by RNAi increased *TaNAC2-5A* expression, resulting in higher chlorophyll content in flag leaves in wheat. On the contrary, the strong silencing of *TaCKX1* led to a lower chlorophyll content and accelerated leaf senescence [58]. The results provide a path forward to find out the possible roles of the *TaNAC2-5A* transcription factor in wheat senescence.

*TaSNAC11-4B* is another GFM of wheat *NAC* TFs, which positively regulates leaf senescence. Its expression is upregulated when wheat plants are treated with ABA and subjected to drought stress. Structural analysis showed that the transcriptional activity of *TaSNAC11-4B* is determined by its C-terminal region [70].

Chapman et al. [60] reported an NAC domain containing novel *NAM-1* alleles. They established that *NAM-1* played a significant role in delayed leaf senescence leading to an increase in grain filling time in two wheat cultivars. In another study, missense mutations inflicted on five highly conserved residues of *NAM-A1* caused peduncle senescence in bread wheat, while no consistent effect was observed on flag leaf senescence in glasshouse conditions. However, a missense mutation led to a consistent delay in peduncle and flag leaf senescence under field conditions. The function of *NAM-A1* suggests that it is more strongly associated with peduncle senescence than flag leaf senescence [71].

Another wheat NAC protein, NAM-B1, which is encoded by Grain Protein Content-B1 (Gpc-B1) has been reported to increase flag leaf senescence. Phylogenetic analysis revealed that the wheat NAM-B1 protein is closely related to many Arabidopsis NAC proteins, such as *ANAC18*, *ANAC25*, and *ANAC56*. In the grain-filling stage, NAM-B1 regulates the distribution of nutrients from the leaves to the ears, therefore improving the senescence of the flag leaves [47]. In a confirmatory experiment, the knockdown of *NAM-B1* (*Gpc-B1*) resulted in delayed leaf senescence along with increased nutrients in flag leaves and lower nutrients in grains. Analysis of the *HvNAM-1* and *HvNAM-2* genes in barley (*Hordeum vulgare*) revealed that they were homologs of wheat *NAM-B1* [47]. Three allelic versions of *NAM-B1* have been classified according to wild, mutated, and deleted types [72]. Another study showed that one of the functional alleles of *NAM-B1* improved the zinc, iron, and protein content of the grain during early senescence and, as a consequence, the grain filling time and grain weight were reduced [73]. Functional divergence of *NAC* orthologs to the *GPC* gene has been reported in wheat and rice. Wheat *GPC-B1* is located on chromosome 6B while its homologue, *GPC-B2,* is located on chromosome 2B. In rice, the *OsNAC10* gene (ID: Os07g37920) is the closest homolog (ortholog) to both wheat *GPC* genes. Both *GPC* genes showed higher levels of transcript in leaves compared to anthers, and the knockout of these genes in wheat caused delayed senescence with normal fertility. However, in contrast to wheat *GPC*, the functional ortholog *OsNAC10* in rice showed a higher transcript level in anthers than in leaves and did not affect senescence. RNAi-silenced rice plants showed a reduced number of viable pollens [74].

Wheat *TaNAC29* has been reported to regulate not only leaf senescence, but also to increase tolerance to various abiotic stresses. Furthermore, the functioning of *TaNAC29* depends on ABA and some antioxidant enzymes [75]. The analysis of the gene regulatory network of wheat leaf senescence, which consisted of at least 61 NAC genes, showed differential expression from early to late leaf senescence [59]. Among them, the function of only a few wheat NAC genes has been reported.

### 5.3. Seed-Associated Traits Regulated by Wheat NACs

Several seed traits, such as seed germination, seed vigor, seed protein content, and endosperm starch content, are associated with wheat yield and quality. Most seed-related traits are regulated by transcription factors. However, considering seed storage proteins (SSPs), very few TFs have been reported. The SSPs in seeds impart elasticity to wheat flour and, due to the different values of this parameter, flour is used for diverse types of food [34,76,77]. The genome-wide study of transcriptional factors that regulate SSP proteins revealed that *Triticum urartu TuNAC74* binds to the *cis*-elements on the promoters of SSP-related genes and increases their activity, while the knockdown of *TaNAC74* in wheat reduces SSPs in seeds by 24%. The overexpressed *TaNAC74* plants also showed a higher germination rate [76]. *T. urartu* is an ‘A’ genome donor for bread wheat and having a simple genome is a model for wheat. *TuSPR* is another gene coding NAC TF from *T. urartu*. Overexpression of this gene reduced the total content of SSP by 15.97%, while knockdown of its homologue (*TaSPR*) in bread wheat increased the total content of SSP from 7.07 to 20.34%. Analysis revealed that TuSPR *cis*-regulated the *SSP* genes by binding to the 5′-CANNTG-3′ sites distributed in the promoter region [3].

Starch is found in abundance in cereal grains, making it one of the most significant constituents of our diet. The starch content determines the seed yield, quality, and endosperm development. Therefore, it is important to understand the mechanisms of starch content regulation to improve seed yield and quality [78,79]. The quality of wheat flour is also determined by high molecular weight glutenin subunits (HMW-GS), which are important components of SSPs. *TaNAC100* overexpression caused a reduced content of HMW-GS, and as a result, total SSPs were also reduced. However, overexpression of this gene increased the expression of two starch synthesis genes, *TaGBSS1* and *TaSUS2*, resulting in a significantly higher seed starch content. Other phenotypic characteristics improved by *TaNAC100* overexpression included seed size and thousand seed weight [34].

In the endosperm of cereal grains, starch synthesis starts after sucrose transport. Many enzymes such as sucrose-synthase, ADP-glucose-pyrophosphorylase, phosphoglucose isomerase, and phosphoglucomutase play their role in converting sucrose to ADP-glucose and their transport to amyloplast with the help of the BRITTLE-1 (BT-1) protein, which acts as an ADP-glucose transporter [79,80]. Numerous TFs that regulate the mechanism of starch synthesis have been reported in cereals, including rice bZIP58, NF-YB1, and NF-YC12 [81,82,83]. Furthermore, *ZmaNAC36* also known as *ZmaNAC130* and *ZmaNAC128* from maize has also been reported to play a role in starch biosynthesis [84,85]. Similarly, *TaNAC19-A1* in wheat negatively regulated starch biosynthesis in grain endosperm. Overexpression of this gene reduced starch content by binding to 5′-ACGCAG-3′*cis*-elements of the promoters of *TaAGPS1-A1* and *TaAGPS1-B1*, which were predominantly involved in starch synthesis [39]. The conserved domains of common grain-related NAC TFs may have common *cis*-elements or binding sites in the promoter regions. However, the difference in negative or positive effects may be due to differences in the C-terminal domain of transcription factors. For example, in wheat, TaNAC019-A1 binds to *5′-ACGCAG/A-3′*. A similar *cis*-element is also present in maize, compatible with two other NAC proteins ZmNAC128 and ZmNAC130, and conversely, in wheat, they positively regulate starch biosynthesis [39,85].

Seed vigor is also a critical trait for crop yield. This trait depends on the timing and uniformity of seed germination. TaNAC2 is a well-established wheat TF that regulates vigor in seeds by binding to the *cis*-element at the promoter of the *TaNRT2.5-3B* gene. This gene participates in the acquisition of nitrates from the soil [69]. Two *NAC* rice genes, namely *OsNAC-20* and *OsNAC-26*, have been reported to regulate seed storage protein and starch [86]. It will be interesting to find the homolog of NAC TFs, already reported from well-established cereal crops, in wheat and perform its functional characterization for crop improvement.

### 5.4. NAC-Dependent Root Modulation in Wheat

Root organs are important for the uptake of water and nutrients from the soil, and the importance of the root system increases when the soil is deficient in water. Like any other crop, in wheat, a well-established root system ensures crop yield and quality [87]. Many TFs have been reported to have a significant effect on the plant root system. *TaRNAC1* is preferably expressed in wheat roots [88]. Overexpression of this gene resulted in an improvement in root length and aboveground biomass. Furthermore, the transgenic plants showed better tolerance to drought under the applied conditions of PEG treatment. *TaRNAC1* has also been reported to interact with *GA3-ox2* and enhance its expression in roots. *GA3-ox2* encodes an enzyme that converts the inactive form of gibberellin to the active one [88]. Furthermore, overexpression of *TaSNAC8-6A* led to the activation of various drought-responsive and auxin-signaling genes and subsequently helped to develop lateral roots [89]. Previously, it has been proven that tae-miR-164 modulates many NAC TFs in wheat; however, in a recent study, tae-miR-164 has been reported to target and down-regulate the *TaNAC14* gene. Experiments with overexpressed tae-miR-164 and *TaNAC14* demonstrated inhibition of root development and reduced tolerance to drought and stress. Therefore, *TaNAC14* negatively regulates root development in wheat crops [90].

As a result of global warming, water evaporation poses a serious threat to water reservoirs and causes droughts [91]. Therefore, the productivity of cereal crops will mainly depend on the ability of the roots to extract deep water from the soil. Several recent studies on the plant root system have shown that longer and deeper roots with increased diameter have increased crop yield and quality under reduced water availability conditions [88,92]. However, there are very limited gene and transcription factor data available that indicate the modulation of root systems in cereal crops, including wheat. Therefore, the identification and characterization of more root-regulating *TaNACs* could be beneficial for wheat improvement. Figure 3 provides a pictorial overview of organ-specific *TaNACs.*

### 5.5. Role of Wheat NACs in Abiotic Stresses

Biotic and abiotic stresses trigger various changes in plants, from transcription to the level of metabolism, which affect plant growth, development, and yield. TFs interact with functional *cis*-elements in the promoter regions of stress-related genes and regulate them to achieve better whole-plant tolerance to various biotic and abiotic stresses by either overexpression or underexpression of the genes. Plant NAC proteins are most widely identified as stress-related; however, initially plant NAC proteins were associated with plant development [11,13,93]. Like other crops, wheat also shows a wide range of NAC genes involved in the regulation of biotic and abiotic stress. *TaNAC2* is associated with multiple abiotic stresses such as drought, salt, and freezing [94]. Transgenic Arabidopsis was found to be stress tolerant when wheat *TaNAC2* was overexpressed [94]. Furthermore, the regulation of *TaNAC2L* has been reported to be heat-dependent and the expression of this gene has increased significantly in response to higher temperatures. Arabidopsis overexpressing *TaNAC2L* showed enhanced tolerance to a higher temperature, allowing transgenic plants to be more thermotolerant [95]. Furthermore, *TaNAC29* overexpression in Arabidopsis allowed transgenic plants to be more tolerant to drought and salt stresses and also showed a hypersensitive response to ABA [75].

The concentration of hydrogen peroxide (H_2_O_2_) is an important indicator of abiotic stress conditions in plants. During abiotic stress, plants have a higher oxidation rate, especially in membrane lipids [96,97]. Treatment with H_2_O_2_ in Arabidopsis increased salt and drought-related transcripts of the *JUB1* and *ATAF1* genes [98,99]. Activities of antioxidant enzymes such as superoxide dismutase (SOD), peroxidase (POD), and catalase (CAT) have also increased during salt and drought, and prolonged stress gradually decreased their activities [96,97]. Transgenic plants overexpressing *TaNAC29* also showed less accumulation of H_2_O_2_ and increased CAT and SOD, leading to increased salt and drought stress. Furthermore, the coexpression analysis indicated that the transcript level of two senescence-associated genes (SAG) *SAG13* and *SAG113* also decreased [75]. *TaNAC47*, another wheat *NAC* gene, is differentially expressed in different tissues of wheat plants in response to the various abiotic stresses. The results suggested that *TaNAC47* activated multiple downstream gene expressions. They also uncovered its function as a transcription factor, as it bound to the ABRE *cis*-element in the yeast one-hybrid assay. In Arabidopsis, *TaNAC47* overexpression increased the tolerance of transgenic plants to abiotic stresses such as cold, salt, polyethylene glycol (PEG), and ABA [100].

Drought is one of the devastating abiotic stresses to cereal crops that causes almost 14% of the average losses worldwide. Wheat is also strongly affected by water scarcity; therefore, drought-tolerant germplasm is urgently needed, and this can be achieved with new wheat breeding techniques [89,101]. Many functions of NAC proteins have been validated by overexpression or knockdown experiments to confirm their role in various stresses, including drought stress. An example is *TaSNAC8-6A*, which belongs to the wheat *NAC* subfamily. Overexpression of this gene in wheat and Arabidopsis increased tolerance to drought in both transgenic plant lines in an auxin-induced response [89]. Overexpression of rice *SNAC1* under the maize ubiquitin promoter leads to improved tolerance to drought and yield in wheat without losing phenotypical traits [89,102,103]. Similarly, transgenic wheat lines overexpressing *TaRNAC1* showed tolerance to drought, increased grain weight, and overall biomass under PEG treatment [88]. *TaNAC69* is another well-characterized transcription factor for tolerance to drought. Transgenic wheat plants that overexpress *TaNAC69* under the drought-specific promoter *HvDhn4* exhibited greater tolerance to drought stress under PEG-induced dehydration conditions [104]. Two consensus sequences, expanding to 23–24 bp, were identified for the DNA binding of *TaNAC69*. The TaNAC69 ortholog was also identified in Arabidopsis [105]. Another *NAC* gene, *TaNAC67*, has been associated with various abiotic stresses such as drought, salinity, and freezing. Like many other *NACs*, *TaNAC67* is also ABA-responsive and activates multiple stress-related genes, namely *DREB2A*, *COR15*, *ABI1*, and *ABI2. DREB2A*. In addition to tolerance to abiotic stresses, transgenic plants also showed a higher chlorophyll content and improved cell membrane strength [106].

Ion flux measurement is a technique that allows for the measurement of salt stress tolerance in plants. NaCl-induced measurement of K^+^ efflux is a recommended technique to measure salt tolerance in wheat, barley, and Arabidopsis, as represented in Figure 4 [107,108,109]. Transgenic plants overexpressing *TaNAC67* showed higher K^+^ and Na^+^ efflux rates while there was no significant effect on the H^+^ ion flux rate [106]. Structural analysis of *TaNAC67* indicated that its DNA binding domain had a high similarity index to other *NACs* such as rice *OSNAC1*, *TaNAC2*, *TaNAC2A*, *TaGRAB1 TaNAC4*, and *TaNAC69*, but differs in its C-terminal domain related to transcriptional activity [106]. Stress-related NAC TFs are denoted as SNAC and there are at least 41 stress-related SNAC TFs that fall into 14 different groups by phylogenic analysis. *TaSNAC4-3D* encodes one of those drought stress-related NAC TFs that negatively regulates drought stress in wheat when induced by ABA, leading to oxidative damage to plant cells [110]. *TaSNAC4-3A* overexpression in Arabidopsis resulted in increased tolerance to drought by regulating stomatal opening [111].

### 5.6. Role of Wheat NACs in Biotic Stress

Plants are widely attacked by pathogens including bacteria, viruses, and fungi throughout their life cycle. After a pathogenic attack, plants generally show two types of innate immune response, pathogenic-triggered immunity (PTI) and second, effector-triggered immunity (ETI) [9]. PTI is the first layer of immunity in plants against all microbial attacks, while EFI is a specific immune response that is activated by the interaction between the R protein of plants and the effector proteins of a pathogen [112,113]. The role of NACs in plants is not limited to abiotic stress. Their functional characterization in biotic stress by overexpression or knockdown experiments in Arabidopsis, rice, wheat, and other plants has been performed in a broad way. Numerous NAC proteins have been reported to play a pivotal role in plant immunity by being positive or negative modulators. Many of them modulate immunity by host hypersensitive response (HR) and stomatal regulation of the pathogen [9]. The HR represents one type of programmed cell death (PCD) that prevents the spreading of the pathogen by triggering a primary immune response [114]. OsANC4 is a rice NAC TF that triggers the HR. It was reported that infection of *Acidovorax avenae* led to enhanced PCD in plants that overexpressed *OsNAC4* [115]. Moreover, Arabidopsis overexpressing *NAC4* also showed enhanced HR-mediated PCD against pathogenic attack [116].

Stomata not only passively regulate pathogen entry but also play an important role in innate immune responses. Their opening and closing by guard cells represent the first encounter during the plant-pathogen interaction. Three Arabidopsis *NACs*, *ANAC19*, *ANAC55*, and *ANAC72*, have been reported to regulate stomatal innate immunity by activating signaling pathways (Figure 5) [9,117]. Since Arabidopsis is considered the basic model plant, it will be important to find orthologues of *ANAC19*, *ANAC55*, and *ANAC72* in wheat to confer stomatal innate immunity.

Like other crops, multiple biotic stresses greatly affect wheat yield around the world. Wheat leaf rust is one of the most serious diseases. At least 186 *TaNAC* transcripts were obtained from stripe rust and powdery mildew resistant wheat [118]. In a recent study, Zhang et al. [119] reported that *TaNAC35* played an important role in the negative regulation of leaf rust resistance against *Puccinia triticina*. *Puccinia striiformis* is another fungal pathogen causing stripe rust in wheat. *TaNAC30* has been reported to be a negative regulator of resistance to this disease [120]. When wheat plants were infected with *P. striiformis*, the expression of *TaNAC30* increased. On the contrary, silencing of *TaNAC30* resulted in increased resistance to rust fungus along with the accumulation of hydrogen peroxide (H_2_O_2_) in wheat cells. Similarly, *TaNAC2* was also documented to be a negative regulator of resistance to stripe rust in wheat plants. Inhibited expression of *TaNAC2* not only reduced *P. striiformis* (*Pst*) hyphal growth but also resulted in the accumulation of H_2_O_2_, leading to increased resistance to strip rust in an early stage of wheat development [121]. Another *NAC*, *TaNAC8*, encodes a protein containing 481 amino acids, and the gene has its orthologue *OsNAC8* in rice. *TaNAC8* is preferably expressed in seeds rather than in flowers and stems. *TaNAC8* expression was induced by the stripe rust-causing fungus, *P. striiformis*, as well as hormone treatment such as ET and methyl-jasmonate; however, ABA and SA did not induce gene expression. The results of the study indicated that *TaNAC8* responded to strip rust infection and some abiotic stresses such as PEG, salinity, and low temperature [122]. Another wheat *NAC*, *TaNAC4*, which encodes 308 amino acid proteins, has been reported to have a similar function to *TaNAC8*. *TaNAC4* is a homoeolog of rice *OsNAC4* and is preferentially expressed in the roots of wheat seedlings compared to leaves and stems. The expression of this gene is induced by the stripe rust pathogen and by the exogenous application of various hormones, including methyl jasmonate, ET, and ABA; however, there was no notable effect of SA. Furthermore, *TaNAC4* has also been reported to regulate various abiotic stresses such as wounding, salinity, and low temperature [123]. Data obtained on *TaNAC*-responsive genes to stripe rust and powdery mildew offered a great source of information on the functions of NAC in resistance to biotic stress in wheat [118].

The expression of many *NAC* genes is regulated by microRNAs. It was proved for *TaNAC21*/22, which was a target gene for tae-miR164 [124]. *TaNAC21/22* has been reported to be a negative regulator of resistance to wheat stripe rust disease. Another disease, powdery mildew, is one of the most notorious diseases of wheat crops around the world, and its causative pathogen is *Blumeria graminis (Bgt).* Three homoeologs of *TaNAC6*, *TaNAC6-A*, *TaNAC6-B*, and *TaNAC6-C* were characterized for their role in resistance to powdery mildew, and each of the homoeologs responded differently to the *Bgt* infection. *TaNAC6A* overexpression increased its resistance to *Bgt* through the JA pathway. In general, *TaNAC6s* played a role in resistance to powdery mildew at the base and the broad spectrum levels [125]. Another wheat disease, *Fusarium graminearum*, is the causative agent of Fusarium Head Blight (FHB) disease. Enhanced resistance to FHB was reported in wheat lines overexpressing *TaNACL-D1*. This gene encoding NAC-like TF, TaNACL-D1, was found to interact with *TaFROG*, encoding an orphan protein. Furthermore, *TaNACL-D1* was proved to be an orthologue of a *Poaceae NAC*; however, the C-terminal of TaNACL-D1 is *Triticeae* specific. Previously, *TaFROG* was reported to regulate signal pathway activation by interaction with *sucrose nonfermenting1 related kinase1* (*SnRK1*) and improve resistance to *Fusarium* head blight disease [126]. Biotic stresses pose serious threats to wheat yield and production, and wheat NACs have great potential to cope with these challenges. Table 1 is a comprehensive summary of wheat NACs and their associated functions.

## 6. Future Prospects and Conclusions

NAC TFs are the key regulators for plant growth and development. The indispensable role of plant NACs in various stress regulations has shown that they are imperative candidates in the development of stress-tolerant crops. Being one of the largest families of plant TFs, NACs play a role in improving grain yield, establishing lateral roots, delayed senescence, and abiotic and biotic resistance by modulating the transcriptional regulatory activities of the gene network. Genome-wide analysis and transcriptome profiling could be helpful in the identification of various yield-related and stress-related NAC TFs under different stress conditions. Most NAC TFs show transcriptional regulatory activity by interaction with *cis*-elements, and very little is known about the dynamics of a stabilized TFs-CEs interaction. Knowledge of CEs that where, when, and what factors activate these regulatory sites distributed in the genome is a challenge in understanding the *cis*-regulatory mechanisms. Another challenge is to find unbiased approaches that could identify active CEs. More studies should be conducted on the role of C-terminal domain variability, since it constantly makes NAC TFs divergent.

Most of the advances that have been made regarding *TaNACs* are in finding their *cis*-elements in promoter regions and their associated function, but almost no work has been carried out to find *cis*-elements that are distributed in proximal and distal locations of the targeted gene. The proximity interactions of *TaNACs* would pave the way for the underlying mechanism of gene regulatory networks. Since climate changes pose serious threats in the form of drought, salt, and heat stress to crop yield and productivity, an exploration of the role of NAC TFs becomes even more crucial. Biotechnological tools, which allow us to obtain overexpression, silenced expression, knockdown, or any edition of a targeted gene, have proved to be handy tools for finding the functions associated with the gene. Knowledge about coregulations of *NACs* with other genes can further explain the gene network cooperation. Although in the wheat crop the 263 *TaNAC* genes have been reported, only a few have been characterized for their role and functions. TaNAC TFs have huge potential that must be exploited to improve wheat. We summarized the knowledge about the investigated *TaNAC* genes and their associated functions, which can be helpful in improving the performance wheat yield. Depending on the positive or negative regulation of yield-related traits, the genes might be overexpressed, down-regulated, or knocked down by biotechnological tools. The most precise and widely used is the CRISPR/Cas9 technology, which can be applied for both, CRISPR/Cas9-mediated knockout or gene activation. For example, *TaNAC-S-A1*, which is a positive regulator of grain yield and chlorophyll content, should be overexpressed to enhance yield potential. Inversely, the precise knockout of *TaNAC19-A1*, a negative regulator of starch biosynthesis in grain endosperm, would be beneficial in increasing the starch content and the same grain weight. The orthologs of *NAC* in rice, barley, and maize, which are known to play an important role in plant development, could be indicators for the future search for new *NACs* of wheat.

## Figures and Tables

**Figure 1 ijms-23-15431-f001:**
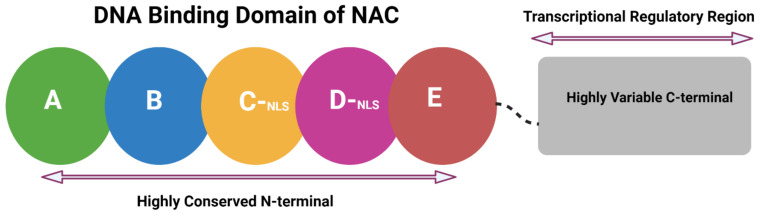
Schematic diagram of a typical NAC TF representing highly conserved N-terminal DNA binding domain (A–E subdomains) and highly variable C-terminal transcriptional regulatory region.

**Figure 2 ijms-23-15431-f002:**
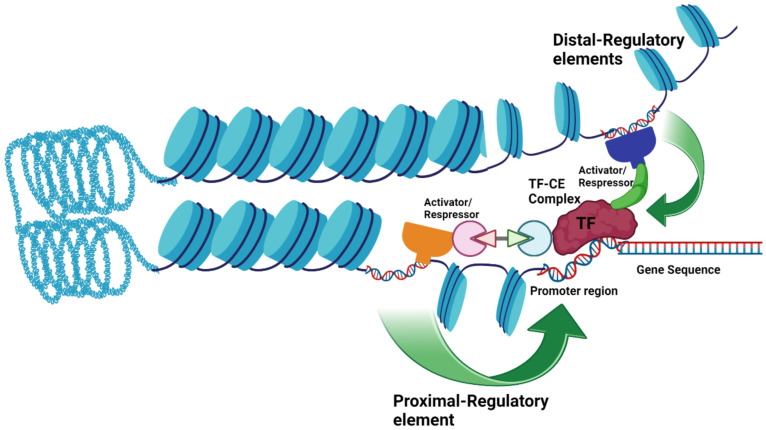
Illustration of *cis*-regulation of a gene: TF (red) can directly bind to CE in the promoter region and regulate gene expression or CEs located in proximity (proximal or distal locations) bind to the different activators or repressors (dark blue, green, light green, pink, orange) and then interact with the core promoter and regulate gene expression.

**Figure 3 ijms-23-15431-f003:**
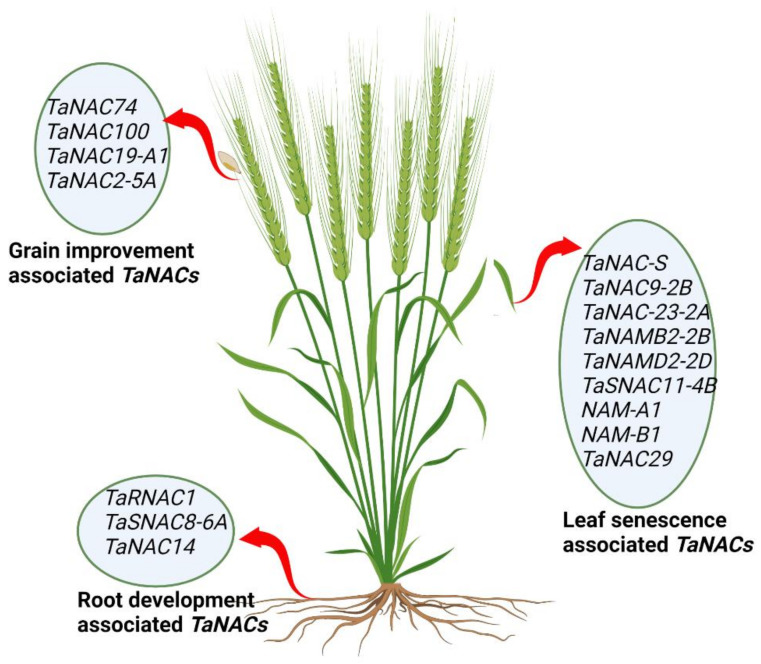
Pictorial representation of *TaNACs* associated with root development, grain improvement, and leaf senescence in the wheat plant.

**Figure 4 ijms-23-15431-f004:**
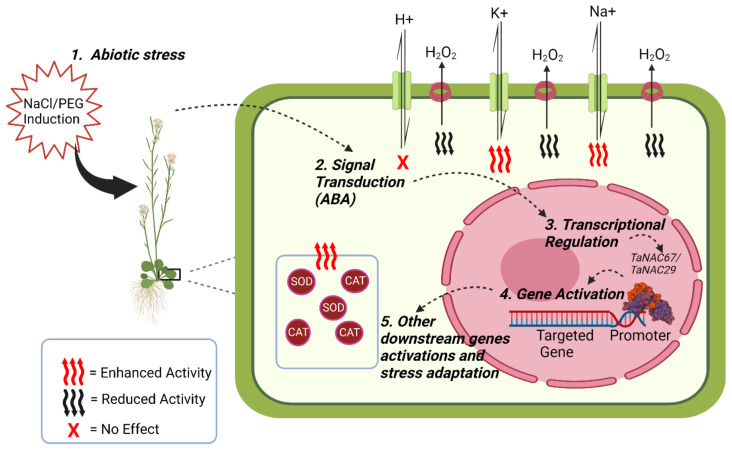
An overview of the regulation of abiotic stress by *TaNAC67* and *TaNAC29* in transgenic Arabidopsis. (1) Transgenic Arabidopsis overexpressing *TaNAC67* and *TaNAC29* acquired tolerance to abiotic stress after NaCl and PEG treatment, respectively. (2) NaCl and PEG-induced abiotic stress tolerance leads to transduction of the ABA signal (3 and 4). Transduction of the ABA signal leads to transcriptional regulation by TaNACs to activate their targeted genes. (5) As a consequence of targeted gene activation, some other downstream genes are activated and lead to general stress management. Transgenics overexpressing *TaNAC29* accumulated less H_2_O_2_ and showed increased CAT and SOD enzyme activities that made transgenics salt and drought tolerant. Similarly, transgenics overexpressing *TaNAC67* had higher efflux rates of K+ and Na+ ions, and no significant effect on the H^+^ ion flux rate allowed transgenics to be more salt-tolerant.

**Figure 5 ijms-23-15431-f005:**
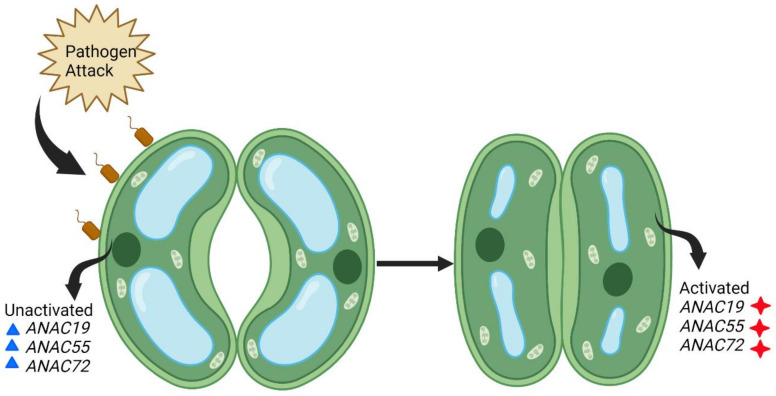
Stomatal innate immunity conferred by three Arabidopsis *NACs* against biotic stress. Pathogen (brown) attack induces activation of the *ANAC19*, *ANAC55*, and *ANAC72* genes (inactivated genes are in blue and activated in red) and signal pathways in Arabidopsis and, as a consequence, the stomata confer innate immunity by closing the guard cells.

**Table 1 ijms-23-15431-t001:** Summary of wheat *TaNAC* genes and their associated functions.

*TaNACs*s	Associated Function	Positive/ Negative (+/−) Regulation	Method of Functional Characterization	Co-Regulation /Interaction /Activation of Other Genes	Cis-Regulatory Sites	References
*TaNAC74*	Seed storage proteins (SSPs), seed germination rate	− regulator of SSPs + regulator of seed germination rate	Overexpression and knockdown	* TaSPR *	5′-CANNTG-3′	[3]
* TaNAC100 *	Seed storage proteins (SSPs), starch, grain size, and weight	− regulator of SSPs *+* regulator of seed starch, seed size, and thousand seed weight	Overexpression	*TaGBSS1* and *TaSUS2*	-	[34]
*TaNAC-S-A1*	Grain yield, chlorophyll contents	*+* regulator of grain yield and chlorophyll contents	Transcriptome analysis	-	-	[59,67]
*TaNAC-S-7B2*	Grain protein contents	+ regulator of grain protein contents	Transcriptome analysis	-	-	[59,67]
* TaNAC19-A1*	Starch synthesis	− regulator of starch biosynthesis in endosperms	Overexpression	*TaAGPS1-A1* and *TaAGPS1-B1*	5′-ACGCAG-3′	[39]
*TaNAC2-5A*	Seeds vigor	+ regulator of seeds vigor	Overexpression	*TaNRT2.5-3B*	-	[69]
*NAM-B1*	Grain protein, leaf senescence	+ regulator of grain protein, leaf senescence	Knockdown	-	-	[47]
*TaNAC-S*	Leaf senescence, grain yield, and grain protein contents	− regulator of leaf senescence + regulator of grain yield and grain protein contents	Overexpression	-	-	[66]
*TaSNAC11-4B*	Leaf senescence	+ regulator of leaf senescence	ABA-induced expression	ABA-pathway responsive	-	[70]
*NAM-1*	Leaf senescence	+ regulator of leaf senescence	Bulk segregant analysis, Missense mutations	-	-	[60,71]
*TaRNAC1*	Roots length, above-ground biomass, drought tolerance	+ regulator of roots length, above-ground biomass, drought tolerance	Overexpression	GA3-ox2	-	[88]
*TaSNAC8-6A*	Lateral roots development	+ regulator of	Overexpression	activate various drought-responsive and auxin-signaling genes	-	[89]
*TaNAC14*	Root development	− regulator of root development	Overexpression	tae-miR-164	-	[90]
*TaNAC29*	Leaf senescence, Drought, and salt stresses	+ regulator of leaf senescence, drought, and salt stress tolerance	Overexpression	CAT and SOD enzyme and ABA-pathway responsive	-	[75]
*TaNAC2*	Drought, salt, and freezing stress	+ regulator of drought, salt, and freezing stress tolerance	Overexpression	-	-	[94]
*TaNAC2L*	Heat	+ regulator of thermotolerance	Overexpression	-	-	[95]
*TaNAC47*	Cold, salt, polyethylene glycol (PEG), and ABA	+ regulator of cold, salt, polyethylene glycol (PEG), and ABA stress tolerance	Overexpression	-	-	[100]
*TaSNAC8-6A*	Drought stress	+ regulator of drought tolerance	Overexpression	ABA-pathway responsive	-	[89]
*TaRNAC1*	Drought stress, grain weight, and biomass	+ regulator of drought tolerance, grain weight, and biomass	Overexpression	PEG pathway responsive		[88]
*TaNAC69*	Drought stress	+ regulator of drought tolerance,	Overexpression	PEG pathway responsive		[104]
*TaNAC67*	Drought, salinity, and freezing stress	+ regulator of Drought, salinity and freezing tolerance, cell membrane stability, cell membrane stability	Overexpression	*DREB2A*, *COR15*, *ABI1 and ABI2. DREB2A*		[106]
*TaSNAC4-3D*	Drought stress	− regulator of drought tolerance	Overexpression	ABA-pathway responsive	-	[111]
*TaNAC35*	Leaf rust stress	− regulator of wheat resistance to leaf rust	Knockdown	-	-	[119]
*TaNAC30*	Strip rust stress	− regulator of wheat resistance to strip rust	Knockdown	ABA-pathway responsive		[120]
*TaNAC2*	Strip rust stress	− regulator of wheat resistance to strip rust	Knockdown	ABA-pathway responsive		[121]
*TaNAC8*	Strip rust stress	− regulator of wheat resistance to strip rust	Ethylene and methyl-jasmonate-induced expression	Ethylene and methyl-jasmonate pathway responsive	-	[122]
*TaNAC4*	Strip rust stress	− regulator of wheat resistance to strip rust	Methyl-jasmonate, ethylene, and ABA-induced expression	Methyl-jasmonate, ethylene, and ABA pathway responsive	-	[123]
*TaNAC21/22*	Strip rust stress	− regulator of wheat resistance to strip rust	Knockdown	tae-miR164	-	[124]
*TaNAC6A*	Powdery mildew stress	+ regulator of powdery mildew resistance	Overexpression	jasmonic acid pathway responsive	-	[125]
*TaNACL-D1*	Fusarium head blight stress	+ regulator of Fusarium head blight resistance	Overexpression	*TaFROG*	-	[126]
*TaNAC2-5A*	Possibly Phytohormonal homeostasis	-	Coregulation	*TaCKX2*	-	[58]

+ positive regulator; − negative regulator.

## Data Availability

Not applicable.

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
