# Peer review of "Cis-Regulation by NACs: A Promising Frontier in Wheat Crop Improvement"

_ijms, 2022, doi:10.3390/ijms232315431_

Round 1
Reviewer 1 Report
Dear authors,
Being one of the largest family of plant TFs, NAC are the key regulators for plant growth and development. NACs play a role in improving grain yield, establishing lateral roots, delayed senescence, and abiotic and biotic resistance by modulating the transcriptional regulatory activities of the gene network. The review manuscript “Cis-Regulation by NACs: a Promising Frontier in Wheat Crop Improvement” provides insight into the cis regulation of genes by wheat NACs for the improvement of yield-related traits, including phytohormonal homeostasis, leaf senescence, seed traits improvement, root modulation, and biotic and abiotic stresses in wheat and other cereals.
However I also found some problems in the manuscript, and I suggest this manuscript can be accepted only after minor revisions throughout the manuscript.
Detailed comments:
1. Please check that the abbreviations throughout the text have the full name at the first occurrence, followed by the abbreviation.
2. P4.L139-157. This paragraph discussion “yield-related traits in wheat and cereals”, but whether NAC transcription factors regulate these traits, such as thousand grain weight, spike number, tiller numbers and spike, are not exemplified by relevant examples. There is also no relevant discussion in the subsequent paragraphs.
3. P6. L229-242. How does NAC regulate to increase yield? Which trait was regulated to make the yield increase?
4. P9. L380-382. Add literature support.
5. P11. L457, 458. effector-triggered immunity (EFI) should be (ETI)
Author Response
Dear Reviewer (1),
Thank you very much for your effort to review our manuscript. Our responses and explanations are under your queries. We believe that you will find them acceptable.
Best regards,
Anna Nadolska-Orczyk
- Please check that the abbreviations throughout the text have the full name at the first occurrence, followed by the abbreviation.
Authors response: The text has been gone throughout and we made the abbreviations followed by the full name in the first occurrences as per suggestion.
- L139-157. This paragraph discussion “yield-related traits in wheat and cereals”, but whether NAC transcription factors regulate these traits, such as thousand grain weight, spike number, tiller numbers and spike, are not exemplified by relevant examples. There is also no relevant discussion in the subsequent paragraphs.
Authors response: The relevant section has been added mentioning the role of wheat NAC genes in yield-related traits improvement from line ------ to --------- as per suggestion. Other, yield-related traits regulated directly or indirectly by NAC TFs are widely described in sections 5.1 to 5.6.
In earlier research TaNACs have been documented to be involved in the regulation of important agronomic traits [46]. Uauy et al. [47] found that Gpc-B1, a quantitative trait locus (QTL) of elevated grain protein, is a NAC TF (annotated as NAM-B1), which also accelerates leaf senescence and increases nutrient remobilization. He et al. [46] found that overexpression of nitrate inducible wheat TaNAC2-5A increased root growth, nitrate uptake rate and grain yield. The allele was also found to bind to the promoter region of genes that encode the nitrate transporter and glutamine synthase, resulting in an improved ability of roots to acquire nitrogen and increased grain yield. Therefore, NAC-encoding genes represent another very important resource for more efficient nitrate use and higher yielding crops [48, 49].
- L229-242. How does NAC regulate to increase yield? Which trait was regulated to make the yield increase?
Authors response: There is a typographical error and the statement misses the word ‘may’.
After correction, the statement says: “The NAC1 type (TaNAC-S) is another type of wheat TF that negatively regulates leaf senescence but interestingly may increase not only grain yield but also grain protein content” because of the increase in the duration of leaf photosynthesis during grain filling. And the similar way original authors of the study concluded their research reference [63] .
- L380-382. Add literature support.
Authors response: The literature support has been added as per suggestion line 382.
- L457, 458. effector-triggered immunity (EFI) should be (ETI).
Authors response: Corrected. Thank you.
Reviewer 2 Report
Dear Authors
The manuscript entitled "Cis-Regulation by NACs: a Promising Frontier in Wheat Crop 2 Improvement" is a comprehensive review of the NAC family of transcription factors. The manuscript content is a comprehensive compilation of different studies. The build-up of the manuscript is also good starting from the basics and then categorizing all the NACs based on their function. However, I feel the manuscript could be improved, and below are few suggestions besides some queries that the author may like to answer and incorporate.
1. Fig. 1 is very general figure and do not display much information rather than saying it has A-E DNA binding domains and highly variable C terminal. It may be deleted or can be displayed in form of conserved sequence of A-E domains.
2. Section 4: Yield-related traits in wheat and cereals: This heading should be Yield-related traits in wheat and emphasize on wheat only. Since the content under this section has been briefly described and not provided sufficient information on other cereals. It will be good if this section can only emphasize on wheat only while describing all the traits in details. Even under this section, there is no need to describe morphology of flower. Possible concentrate on those traits which could be improved by NAC manipulations
3. If possible make a figure showing genomic location of all the TaNACs on the wheat genome.
4. Can this manuscript conclude about some important TaNACs which can be emphasized for wheat improvement.
5. Figure 5 shows the innate immunity conferred by Arabidopsis NACs to biotic stresses. Whether orthologues of these as TaNACs should be the important point of study? If you think, make it a point in figure or text. If not, then this could not be the figure for this manuscript i belive since the manuscript is on role of NACs in wheat.
6. Conclusion of this manuscript may be improved by concluding how these NACs can be manipulated for enhancing yield potential of wheat.
Thanks
Author Response
Dear reviewer (2),
Thank you very much for your time and effort in reviewing the manuscript. We received your review when we had already corrected the manuscript according to the comments of the two previous reviewers. Please find below our answers to your queries. We believe that you will find them acceptable.
Best regards,
Anna Nadolska-Orczyk
- Fig. 1 is very general figure and do not display much information rather than saying it has A-E DNA binding domains and highly variable C terminal. It may be deleted or can be displayed in form of conserved sequence of A-E domains.
Author Response: Fig. 1 belongs to this basic part of the manuscript. The purpose of this figure is to give simple and general idea about NAC proteins, their conserved domains, domains with the nuclear localization signal and variable region. It will be easily remembered by those less familiar with NAC TF readers. Two previous reviewers do not question this figure, so we would like to live it as it is. Some recent studies on NACs such as Liu et 2022 ref. [18] and Singh et al 2021 ref [13] have also represented NACs in a similar way.
- Section 4: Yield-related traits in wheat and cereals: This heading should be Yield-related traits in wheat and emphasize on wheat only. Since the content under this section has been briefly described and not provided sufficient information on other cereals. It will be good if this section can only emphasize on wheat only while describing all the traits in details. Even under this section, there is no need to describe morphology of flower. Possible concentrate on those traits which could be improved by NAC manipulations
Authors Response: We mentioned in the introduction that because of very limited knowledge about NACs in wheat, we are going to compare research in this species with model cereal, which is rice as well as more closely related to wheat barley (both last belong to the Triticeae family). Therefore, a short comparison of flower morphology in these species is important. As we mentioned in the chapter: ‘The overall yield and grain numbers in wheat, barley and rice crops depend on many factors, such as inflorescence morphology, tiller numbers, differentiation, vegetative and reproductive phase time, spike and spikelet initiation, elongation, and maturation.’ The contents of this section was not questioned by two previous reviewers. According to a suggestion of one of them we added to this chapter earlier research documenting TaNASs involved in the regulation of important agronomic traits.
- If possible make a figure showing genomic location of all the TaNACs on the wheat genome.
Authors Response: There are 263 TaNACs that have been reported in wheat and only part of them have been characterized. So, it would not be possible to draw such a figure at this point.
- Can this manuscript conclude about some important TaNACs which can be emphasized for wheat improvement.
Authors Response: We added to the introduction: ‘More and more research indicates that NAC genes are important regulators of yield-related traits. It refers to direct regulation of yield parameters such as seed-associated traits. However, they might be also involved indirectly in yield improvement by regulation of phytohormonal homeostasis, especially in generative organs, root development, leaf senescence and/or biotic and abiotic stresses. All these topics are reviewed and discussed.’
According to your suggestions 4 and 6 more conclusions are included in the ‘Future Prospects and Conclusions’ chapter.
- Figure 5 shows the innate immunity conferred by Arabidopsis NACs to biotic stresses. Whether orthologues of these as TaNACs should be the important point of study? If you think, make it a point in figure or text. If not, then this could not be the figure for this manuscript i belive since the manuscript is on role of NACs in wheat.
Authors Response: Arabidopsis is considered the basic model plant for the study of plant NACs in other plant species, including cereals. Therefore, it will be important to find orthologues of ANAC19, ANAC55, and ANAC72 in wheat to confer stomatal innate immunity, because so far no such study has been carried out. This point has been supplemented in the text as suggested (lines 504 to 505).
- Conclusion of this manuscript may be improved by concluding how these NACs can be manipulated for enhancing yield potential of wheat.
The author’s response is included in the manuscript (page 16): ‘We summarized the knowledge about the investigated TaNAC genes and their associated functions, which can be helpful in improving the performance of wheat yield. Depending on the positive or negative regulation of yield-related traits, the genes might be overexpressed, down-regulated, or knocked down by biotechnological tools. The most precise and widely used is CRISPR/Cas9 technology, which can be applied for both CRISPR/Cas9-mediated knockout or gene activation. For example, TaNAC-S-A1, which is a positive regulator of grain yield and chlorophyll content, should be overexpressed to enhance yield potential. Inversely, the precise knockout of TaNAC19-A1, a negative regulator of starch biosynthesis in grain endosperm, would be beneficial in increasing the starch content and the same grain weight.‘
Reviewer 3 Report
minor revision
This review by Adnan et al., provides insights into the cis regulation of genes by wheat NACs. It summarizes the research progress of TaNACs through the two parts: the structural attributes and functional features. In the functional features of TaNACs section, the functions of TaNACs are described in detail, including phytohormonal homeostasis, leaf senescence, seed traits improvement, root modulation, and biotic and abiotic stresses in wheat and other cereals. It is helpful to further understand the regulatory mechanism of TaNACs transcription factor and provides a basis for genetic improvement of wheat.
In general, this review will contribute to this field. Some suggestions need to be consider to improve the review.
1. It would be better if the TaNACs evolutionary relationship with other species is added in the review.
2. Some NAC transcription factors have been reported as the secondary cell wall biosynthesis master switch. Is there a similar study on NAC in wheat?
3. NACs transcription factors also been reported to participant in biotic and abiotic stresses by binding some cis elements. Do these have common identity?
4. What is the NAC binding site in wheat? Can the author focus on the Cis-regulation of NAC (Line 89)
5. It needs to be briefly introduced in the introduction about “Yield-related Traits in Wheat and Cereals” (Line 139).
Author Response
Dear Reviewer (3),
Thank you very much for your helpful review that improved the manuscript. We made point-by-point corrections in the manuscript, and our responses are provided below your comments.
Best regards,
Anna Nadolska-Orczyk
- It would be better if the TaNACs evolutionary relationship with other species is added in the review.
Authors response: We have tried to mention some of important wheat orthologues in other crops wherever to establish the evolutionary relationship with other species, such as in lines:
299 – 300: Analysis of the HvNAM-1 and HvNAM-2 genes in barley (Hordeum vulgare) revealed that they were homologs of wheat NAM-B1.
Lines 307 – 308: In rice, the OsNAC10 gene (ID: Os07g37920) is the closest homolog (ortholog) to both wheat GPC genes.
Lines 523 – 525: Another NAC, TaNAC8, encodes a protein containing 481 amino acids, and the gene has its orthologue OsNAC8 in rice. TaNAC8 is preferably expressed in seeds rather than in flowers and stems.
Lines 531 – 533: TaNAC4 is a homoeolog of rice OsNAC4 and is preferentially expressed in the roots of wheat seedlings compared to leaves and stems.
However due to manuscript limit and scop constrained to us to mention only few important evolutionary relationships.
- Some NAC transcription factors have been reported as the secondary cell wall biosynthesis master switch. Is there a similar study on NAC in wheat?
Authors response: We have gone through literature review. To our knowledge, there is no study available in wheat right now that mentions the role of NAC in the secondary cell wall switch. However, similar studies have been reported on other crops such as Arabidopsis (AtSND2) and rice (OsSND2).
It will be interesting to find the orthologues of these genes in the wheat crop to characterize the role of wheat NAC in secondary cell wall biosynthesis. This idea has been rightly mentioned in our conclusion.
- NACs transcription factors also been reported to participant in biotic and abiotic stresses by binding some cis elements. Do these have common identity?
Authors response: The conserved domains of common NAC TFs may have common cis-elements or binding sites in the promoter regions. However, the difference in negative or positive effects may be due to differences in the C-terminal domain of transcription factors. For example, in wheat, TaNAC019-A1 binds to 5’-ACGCAG/A-3’. A similar cis-element is also present in maize, compatible with two other NAC proteins ZmNAC128 and ZmNAC130, and conversely, in wheat, they positively regulate starch biosynthesis [83,84].
This could be true for NACs regulating abiotic and biotic stress.
- What is the NAC binding site in wheat? Can the author focus on the Cis-regulation of NAC (Line 89)
Authors response: Cis-regulation of NAC part has been added as per suggestion to page 4.
TaSPR gene from bread wheat encodes NAC protein and it binds to the 5’-CANNTG-3’ CEs distributed in the promoter regions of SSP genes encoding seed storage proteins [3]. Similarly, TaNAC19-A1 NAC protein binds to the 5’-ACGCAG-3’ CEs in the promoter regions of TaAGPS1-A1 and TaAGPS1-B1 [39]. Recent studies regarding TaNACs only focus on finding their CEs distributed in the promoter regions of their targeted genes; however, there is a huge gap in knowledge in finding the CEs in proximal and distal locations of the targeted gene. Exploration of steric stabilization of TaNACs to their CEs is another neglected area of research, which needs to be further focused.
- It needs to be briefly introduced in the introduction about “Yield-related Traits in Wheat and Cereals” (Line 139).
Authors response: Thank you. The text (below) is added to page 2.
More and more research indicates that NAC genes are important regulators of yield-related traits. It refers to direct regulation of yield parameters such as seed-associated traits. However, they might be also involved indirectly in yield improvement by regulation of phytohormonal homeostasis, especially in generative organs, root development, leaf senescence and/or biotic and abiotic stresses. All these topics are reviewed and discussed.